# And Yet They Differ: Reconsiderations of Diversity within *Dactylochelifer latreillii* (Arachnida: Pseudoscorpiones)

Christoph Muster [1,*], Jan Korba [2], Petr Bogusch [3], Petr Heneberg [4] and František Šťáhlavský [2,*]

1   Zoological Institute and Museum, University of Greifswald, Loitzer Str. 26, D-17489 Greifswald, Germany
2   Department of Zoology, Faculty of Science, Charles University, Viničná 7, CZ-128 44 Prague, Czech Republic; korbaja@natur.cuni.cz
3   Department of Biology, Faculty of Science, University of Hradec Králové, Rokitanského 62, CZ-500 03 Hradec Králové, Czech Republic; bogusch.petr@gmail.com
4   Third Faculty of Medicine, Charles University in Prague, Ruská 87, CZ-100 00 Prague, Czech Republic; petr.heneberg@lf3.cuni.cz
*   Correspondence: cm@christoph-muster.de (C.M.); stahlf@natur.cuni.cz (F.Š.)

**Abstract:** Morphological stasis is a widespread characteristic of pseudoscorpions, suggesting that much cryptic diversity remains unexplored. Here, we revise the polytypic species *Dactylochelifer latreillii* in the framework of an integrative taxonomic approach, using DNA barcoding, multivariate ratio analysis, geometric morphometry of the male foretarsus, and genitalic morphology. The pattern of mitochondrial variation suggests three species-level entities in central Europe, which widely overlap in morphospace, but differentiate in the structure of the female genitalia, and by their ecology. *Dactylochelifer latreillii* (Leach) is a halobiont species, occurring exclusively in coastal habitats and in Pannonian salt steppes, and *D. l. septentrionalis* Beier **syn. nov.** is a junior synonym of the nominate species. *Dactylochelifer degeerii* (C. L. Koch) **stat. rev.** is the oldest available name for an inland species that has long been mistaken for the nominotypical subspecies of *D. latreillii*. New habitat information suggests a preference for higher shrub vegetation. *Dactylochelifer ninnii* (Canestrinii) **stat. rev.** is a halophilic Mediterranean species that extends to the northern limits of the Pannonian basin. The distinctiveness of the Mediterranean "form" was recognized by early naturalists in the 19th century, but was ignored by later authorities in the field.

**Keywords:** Canestrini; Cheliferidae; cryptic species; DNA barcoding; geometric morphometry; habitat segregation; median cribriform plate; morphological stasis; multivariate ratio analysis; taxonomy

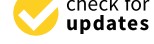



## 1. Introduction

The comprehension and delimitation of species has changed fundamentally with general concepts in biological sciences [1,2]. In Linnean times, species were considered immutable, and thus a typological species concept prevailed [3]. Depending on the degree of phenetic difference, natural variations were stamped with binary nomenclature, resulting in the description of many species that were merely individual aberrations, colour variants or local adaptations. With a time delay for the acceptance of Darwin's evolutionary theory [4], the biological species concept [5,6] overwhelmed the debate in the middle of the 20th century and impacted taxonomic practice [7,8]. With the focus on intraspecific variation, a period of excessive lumping started, in which a plethora of taxa were synonymized with older names. Advances in molecular taxonomy and integrative taxonomy approaches [9–11], however, have frequently resulted in the disproving of "polymorphic" species and in the reinstallation of original species. It has been shown that in certain taxa the species hypotheses proposed by early naturalists of the 19th century correspond better to evolutionary entities than the view of subsequent revisionary authors [12]. This could be attributed to careful and time-consuming observation in the natural habitats, which

helped to distinguish random from evolutionary significant variation. Here, we provide an intriguing example from the arachnid order Pseudoscorpiones.

Pseudoscorpiones is an arachnid order characterized by slow and conservative morphological evolution. Ever since mitochondrial DNA sequences have been generated for pseudoscorpions, strong discrepancies between extensive molecular divergence and the absence of appreciable morphological differentiation have been observed [13]. The mechanisms that cause the morphological crypsis in pseudoscorpions are not fully understood. Principally, three processes may lead to morphological crypsis [14]: (i) recent divergence, i.e., time since speciation was too short to allow morphological differentiation, (ii) morphological convergence, i.e., morphological similarity evolved independently among distantly related taxa in response to similar selection pressures, or (iii) morphological stasis, i.e., niche evolution and hence morphological differentiation across descendant species was constrained by selection (also known as the phylogenetic niche conservatism hypothesis). Christophoryová et al. [15] estimated that cryptic *Lamprochernes* species diverged during the Oligocene era (approximately 31 Ma) and thus concluded that morphological stasis was the effective course. Morphological uniformity can challenge species delineation even if evolutionary distances are large, and traditional species diagnoses based on morphological characters may fail in such groups [16]. There is growing evidence that even in central Europe, a region with a long tradition of faunistic and taxonomic research [17], the true species diversity of pseudoscorpions is not even roughly known. Studies involving DNA barcoding have demonstrated that currently accepted species often constitute complexes of cryptic species [15,18–20]. This was most obvious in non-vagile, soil-dwelling representatives (Chthoniidae, Neobisiidae), with *Neobisium carcinoides* (Hermann, 1804) being the most prominent example. Within this "polymorphic species" [21], at least 12 species-level lineages have been identified [18]. On the other hand, in taxa that disperse phoretically (i.e., transport attached to host species) over long distances (Chernetidae, Cheliferidae) [22], initial studies have suggested little genetic structure and better correspondence with traditional taxonomy [23,24]. Recent studies with geographically more inclusive sampling, however, revealed examples of cryptic diversity in these families as well [25].

*Dactylochelifer latreillii* (Leach, 1817), belonging to the family Cheliferidae, is an exceptionally well known pseudoscorpion species. The World Pseudoscorpiones Catalog [26] lists more than 250 references for this taxon (including its synonyms and subspecies). We are well informed about various aspects of its biology, e.g., the complex mating behavior [27–29], the formation of spermatophores and the sperm transfer [28,30], embryonic [31] and postembryonic development [32,33], the feeding behavior [34], the anatomy of the eyes [35], the gut [36] and the genitalia [37], and the external morphology [38]. On the other hand, taxonomy has remained controversial since the description of *Dactylochelifer latreillii* more than 200 years ago. In the 19th century, seven species were described that are currently considered junior synonyms of *Dactylochelifer latereillii* [26]. Synonymization was pushed forward by the authority of Simon [39,40], but at least with respect to the Mediterranean fauna, doubts remained. For example, Pavesi stated in 1885 "Il sig. Simon dice d'essersi acquistata la certezza che il C. De Géeri C. L. Koch [...] non differisca dal Schaefferi; a me sembrano alquanto diversi, ad ogni modo gli esemplari tunisini sono del tipo Schaefferi" [41]. Also, Ellingsen [42,43] consequently distinguished a Mediterranean form from the typical form, which "must be sought in England". In the 1930s, Beier [44,45] reshuffled the system by proposing a polytypic species with three subspecies: *D. latreillii latreilli* from central and southern Europe, *D. latreillii septentrionalis* from northern Europe, and *D. latreillii cephalonicus* from the Ionian Islands (Greece). The distinction was predominantly based on the dimension and shape of the tarsus of the first leg of the male and its claws, and the proportion of the palpal femur (with respect to *cephalonicus*). Gabbutt [38] provided a first rigorous account on morphometric variation by measuring a large series of British specimens. He found substantial variation in the ratios used by Beier, comprising the ranges for all three subspecies, and suspected that only single specimens were measured by Beier. Mahnert [46] adopted the view of a highly variable species and partly referred

to Gabbutt's study [38] in the synonymization of *cephalonicus* with *latreillii*. Conversely, in support of Beier's subspecies concept, Tooren [47] reported significant differences in the length/depth ratio of the male first pedal tarsus between the coastal form (*septentrionalis*) and the inland form (*latreillii*) in the Netherlands. Differences in the claw length between the subspecies were not corroborated.

The present study was inspired by the coincidence of two occurrences: first, the detection of three deeply diverged COI (cytochrome c oxidase subunit I) lineages within the *Dactylochelifer latreillii* morphotype [25], and second, a field survey on behalf of the German Red List Centre in order to clarify the status of *D. latreillii septentrionalis* in Germany. Through dense sampling in northern Germany, the distribution pattern could be refined and new insights in habitat preferences were obtained. Through the application of an integrative taxonomy approach using DNA barcodes, morphometrics, geometric morphology, traditional (character-based) morphology and natural history, we redefined and characterized three species currently standing as *Dactylochelifer latreillii*.

## 2. Materials and Methods

### 2.1. Taxon Sampling

We reanalyzed and extended the dataset from Just et al. [25] with respect to the "*Dactylochelifer latreillii* morphotype". Field surveys along the German coast and in the northeast German plain were carried out in 2022–2023 for material enrichment and to reveal local distribution and habitat affinities. Fresh material from Great Britain (Camber Sands, East Sussex) was kindly provided by Gerald Legg. A total of 399 specimens from 18 populations were freshly collected (Figure 1, Table S1). Newly collected material was stored in the collections of the Department of Zoology, Faculty of Science, Charles University, Prague (CUNI) and in the collection Christoph Muster, Putbus, Germany (CCM).

Museum material (including types) was examined from the following institutions: Museum für Naturkunde, Berlin, Germany (ZMHB), Naturhistorisches Museum, Wien, Austria (NHMW), Museo Friulano di Storia Naturale, Udine, Italy (MFSN) and from the collection Giulio Gardini, Genoa, Italy (CGG) (Table S2).

### 2.2. Laboratory Procedures and Phylogenetic Reconstruction

The procedure followed, with minor variations, the methodology used in Just et al. [25] to obtain the most comparable results. Whole genomic DNA was extracted from whole specimens using the Genomic DNA Extraction Kit (Tissue) (Geneaid, Taiwan) following the manufacturer's protocol. A fragment of COI was amplified via PCR using primers C1-J-1490/C1-N-2198 [48]. The PCR products were purified using a Gel/PCR DNA Fragments Extraction Kit (Geneaid, Taiwan) and sequenced by Macrogen Europe B.V. (The Netherlands). The obtained chromatograms were assembled, edited, and aligned in Geneious Prime 2022.1.1. Alignment length was set to 555 bp. The obtained alignment ("Dactylochelifer_total") contained 170 sequences (including outgroups) (Alignment S1), 81 ingroup sequences were taken from Just et al. [25] (including one from [49]), and 88 were newly generated. Finally, we obtained a reduced matrix ("Dactylochelifer_haplo") of unique haplotypes (60 sequences, including outgroups) using TCS [50]. Maximum likelihood (ML) analysis was conducted for this reduced alignment with MEGA 11 [51]. For the ML searches, an HKY+G substitution model was applied according to the best Bayesian Information Criterion score and branch support estimated from 1000 bootstrap pseudoreplications. Haplotype networks were constructed with the TCS algorithm using PopART version 1.7 (www.popart.otago.ac.nz, accessed on 23 November 2023).

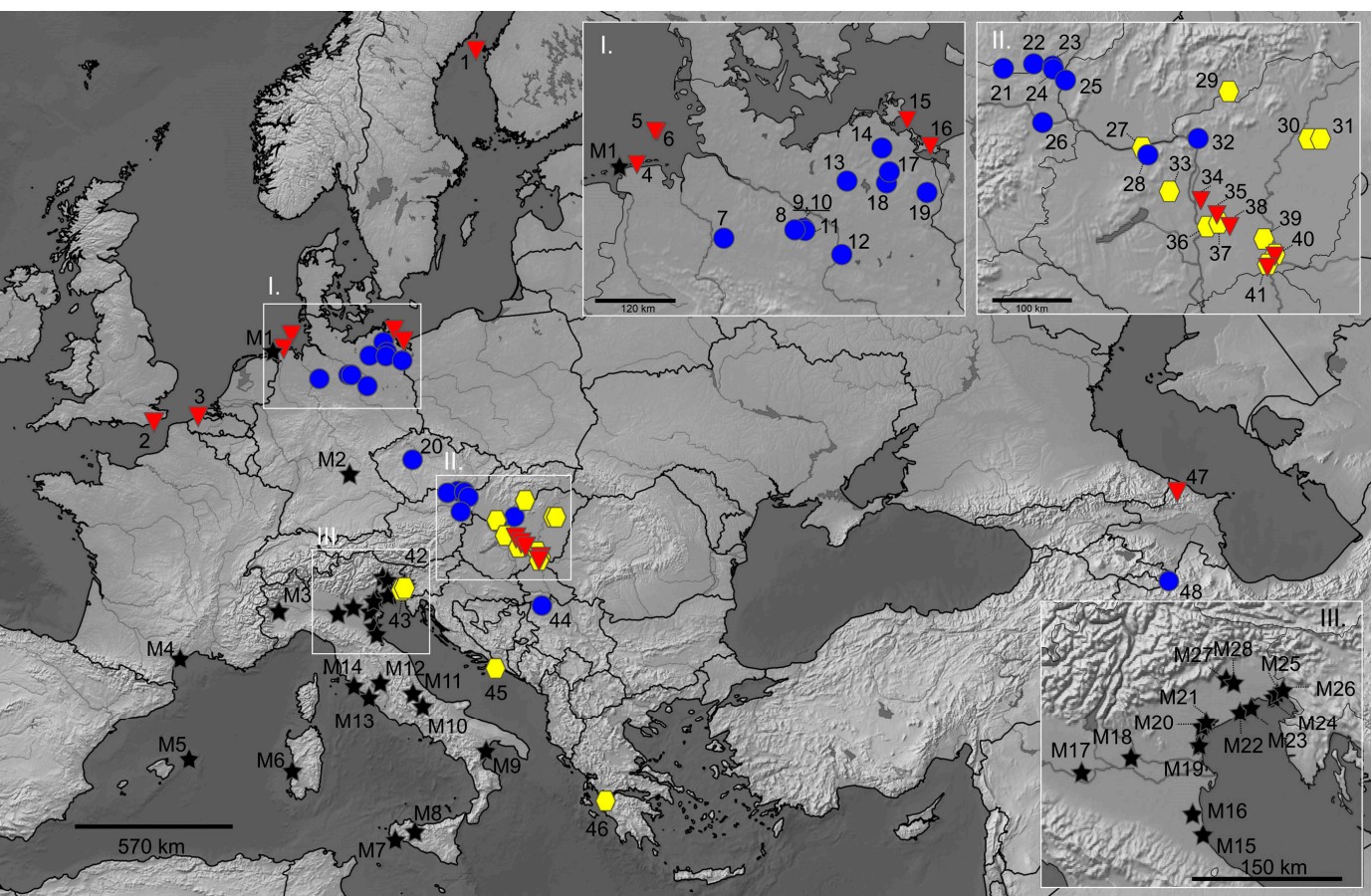

**Figure 1.** Map showing sampling locations of the *Dactylochelifer latreillii* species complex, including data from public databases, analyzed in this paper (No. 1–48, Table S3). The upper left insert (**I.**) shows the sampling locations in northern Germany; the upper right insert (**II.**) shows sampling locations in central Europe, and the lower right insert (**III.**) shows locations in northeastern Italy in detail. Different symbols correspond to different *Dactylochelifer latreillii* lineages according to Just et al. [25]: yellow hexagon: lineage F; red triangle: lineage G; and blue circle: lineage H. The black stars display localities of the museum material observed only morphologically (No. M1-M28, Table S2). All maps were created with the help of an online version of SimpleMappr (https://www.simplemappr.net/).

## 2.3. Morphometric Analysis

Morphometric data were analyzed in the framework of multivariate ratio analysis (MRA) [52] as described by Baur and Leuenberger [53]. MRA provides a set of algorithms based on principal component analysis (PCA) and linear discriminant analysis (LDA) that were specifically adapted for the accurate analysis of body ratios and their interpretation with respect to size and allometric variation. Data were analyzed in the R environment (R Core Team, 2023) using R-scripts and tutorials from the Zenodo platform [54]. First, we calculated isometric size as the geometric mean of all variables. Shape PCR was performed to explore the pattern of overall morphological variation in the dataset and the correspondence with DNA lineages. We plotted isosize against shape PC1 in order to determine the correlation of size with shape, i.e., allometric effects. The LDA ratio extractor was used to extract the best ratios for morphometric discrimination between the three lineages (species) as identified by DNA data. The LDA ratio extractor is a very useful tool for practical taxonomy, as it allows the straightforward interpretation of complex LDR results in terms of body proportions [53]. The tool determines, firstly, the ratio with the largest discriminating power, and then further ratios are extracted that have maximal discriminating power but at the same are as little correlated as possible to the previously selected ratios. For the four best discriminating ratios we calculated

standard distances ($D_{bij}$) and the parameter $\delta$ as a measure of how well size discriminates in comparison with shape. Following the recommendation by the authors [54], we used the LDR extractor in pairwise comparisons between the lineages/species. As there is pronounced sexual dimorphism in pseudoscorpions, the sexes were analyzed separately. Overlap in the morphospace of pairwise comparisons was calculated with the function *Overlap* in the "shipunov" R package [55]. The input matrix contained 34 quantitative measurements taken from 221 specimens (Table S3). Total length was not included in the analyses, as this character is largely dependent on feeding and the desiccation status of the individuals. All measurements were taken according to Figure S1 from digital images using software ImageJ 1.53e (http://imagej.nih.gov/ij, accessed on 19 November 2020). To prevent any possible distortions, we obtained the photographs using a stereomicroscope Olympus SZX12 with camera Olympus DP70 before DNA was extracted from specimens. Ultrastructural differences were compared with scanning electron microscopy (SEM) at the Laboratory of Confocal and Fluorescence Microscope, Faculty of Science, Charles University. SEM photographs were taken with a JEOL JSM-6380LV microscope.

### 2.4. Geometric Morphometrics

The shape of the male tarsus of leg I was analyzed in the framework of geometric morphometrics (GM). In total, 82 specimens were included: 23 specimens from lineage F, 23 from lineage H and 36 from lineage G (Table S3). Photographs of the male foretarsus were obtained as described above. Two fixed landmarks were placed on the apical tip of the tarsi and the ventral joint of the tarsi and contours were captured by 50 semi-landmark points placed equidistantly between them (Figure S2). Landmarks and semi-landmarks were digitized in TPSutil ver.1.81 and TPSdig2 ver.2.32 [56]. Digitized files were analyzed using the "geomorph" package [57] in the R environment (R Core Team, 2023). To eliminate the non-shape-related variation, datasets were subjected to Generalized Procrustes Analysis [58] using the *gpagen* function, and the coordinates of each dataset were used for subsequent analyses of shape variation. The *gm.prcomp* function was used to conduct Principal Component Analysis, and the first two components were utilized to show the shape variation. Procrustes ANOVA using distributions generated from a resampling based on 1000 permutations was employed to test for significance in shape differences between mitochondrial clades using the function *procD.lm* [59]. The generated plots were edited using the programme Inkscape [60] for aesthetic purposes.

### 2.5. Genitalic Morphology

For the examination of internal genitalia, specimens were dissected and the genitalic region was temporarily mounted in Hoyer's solution [61].

## 3. Results

### 3.1. Habitat Segregation in Northern Germany

In northern Germany, a clear pattern of segregated occurrence in two distinct environments emerged. On the one hand, *Dactylochelifer* specimens were collected in sand dunes in close proximity to the coastline. All records arose from tussocks of marram grass (*Ammophila arenaria*), where the species prefers dense stands with old and rotting plant material (Figure 2b). If the specific habitat requirements are met, the species can reliably be found along the entire German coast, including the East and North Frisian Islands, the deep-sea island Helgoland, and the Baltic Sea islands Rügen and Usedom. These are the first records from the Baltic Sea coast in Mecklenburg–Western Pomerania. Inland, on the one hand, *Dactylochelifer* specimens were beaten with great constancy from bushes on humid and nitrogen-rich soils as occurring along floodplains of rivers and rivulets. They showed a clear preference for bushes that were overgrown with climbing plants, primarily hop (*Humulus lupulus*). The highest densities were observed in thickets of dead dry tendrils from previous years (Figure 2c). Elder (*Sambucus nigra*) appears to be another good indicator for the occurrence of *Dactylochelifer* pseudoscorpions. For this habitat type,

the closest record to the sea was approximately 30 km from the coastline. Targeted searches in similar *Humulus*-overgrown bushes close to the sea, for example on the islands of Rügen and Usedom, did not yield any specimens. We also failed to record *Dactylochelifer* in suitable habitats in Western Pomerania east of the river Oder (Poland).

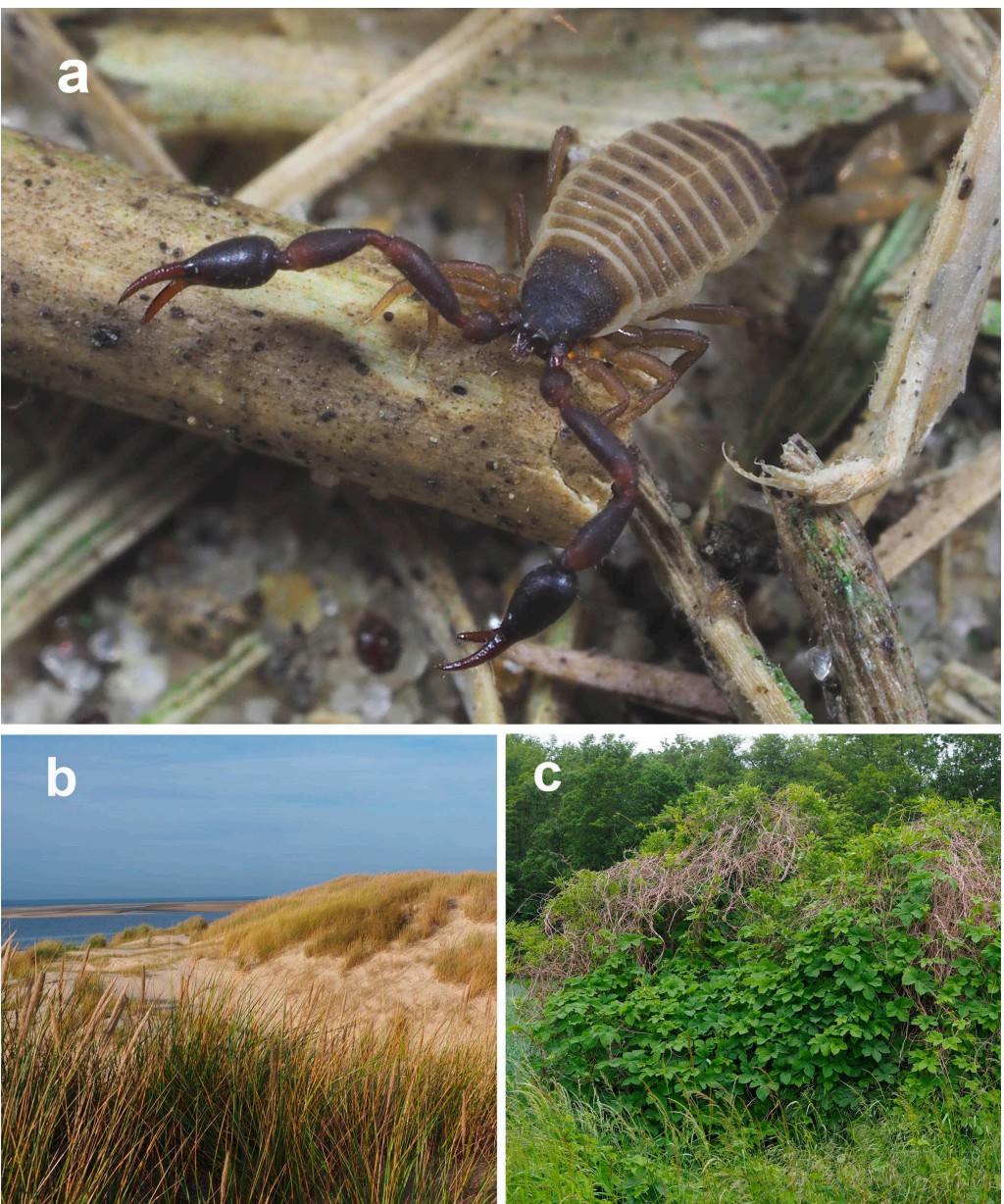

**Figure 2.** (**a**) Female *Dactylochelifer latreillii* from dunes near Ahlbeck, Usedom; (**b**) *Ammophila* dunes at Langeoog, habitat of *Dactylochelifer latreillii*; (**c**) Riparian bushes with *Humulus*, habitat of *Dactylochelifer degeerii*.

*3.2. Molecular Phylogeny and Mitochondrial Divergence*

The maximum likelihood analysis recognized three deeply diverged clades, which for consistency with Just et al. [25] are hereafter referred to as lineages F, G, and H (Figure 3). The phylogenetic tree was rooted with a sequence of *Dactylochelifer copiosus* from GenBank (KT354334). Lineages F and G were highly supported (bootstrap values = 99), whereas lineage H was identified with a bootstrap value of 81. Mean distances between lineages (p-distance, F-G: 11.7%, F-H: 11.3%, G-H: 9.8%) were more than one order of magnitude higher than distances within lineages (F: 1.0%, G: 0.5%, H: 0.7%), thus clearly confirming the

existence of a barcode gap. The distances between the lineages corresponded well to known divergences among closely related species in pseudoscorpions [18] (Figures 2 and 3).

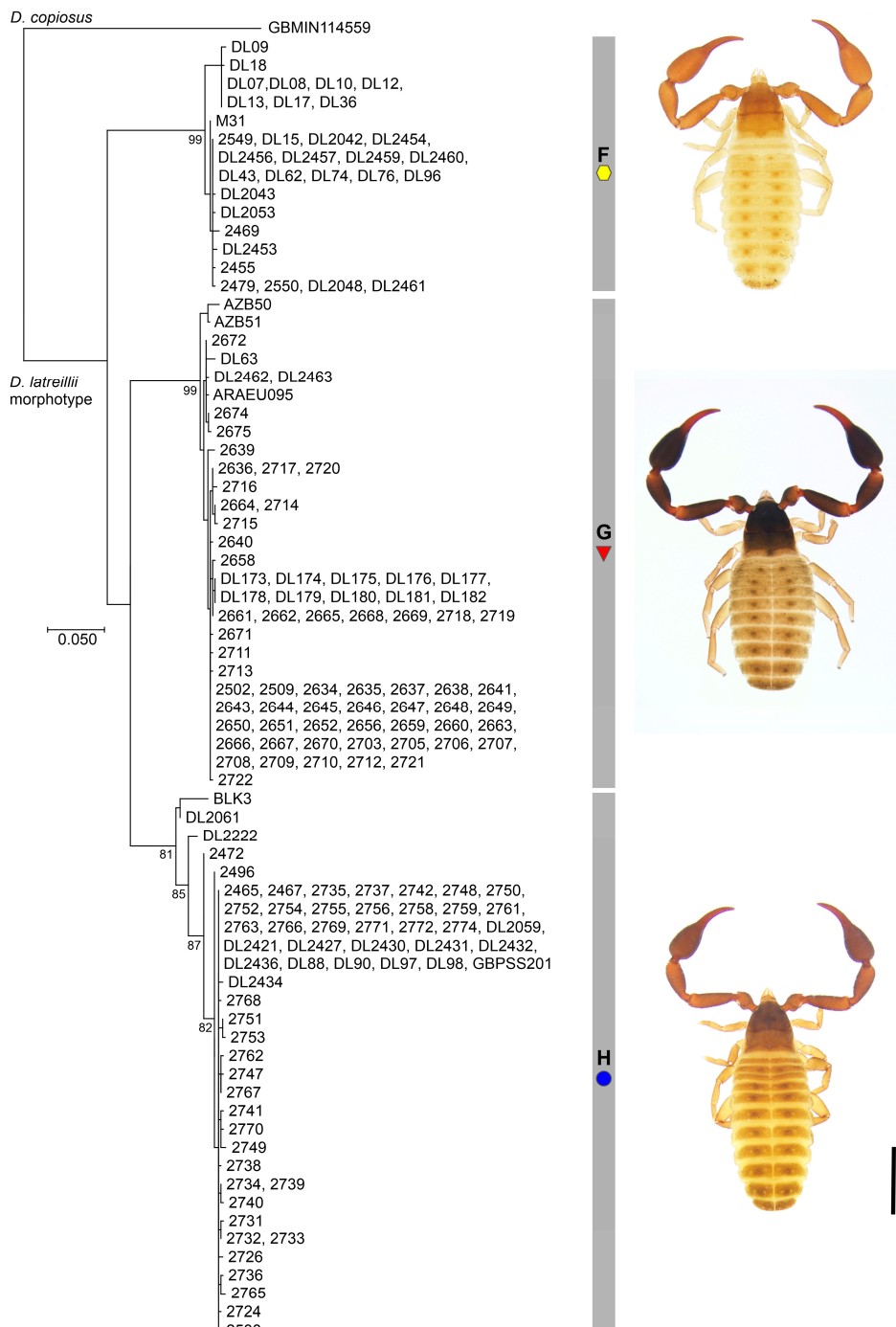

**Figure 3.** The maximum likelihood tree for the *Dactylochelifer latreillii* morphotype based on COI sequences (bootstrap support marked > 80). Designation of the lineages corresponds to Just et al. [25]. Illustrations show typical females of each lineage. Scale bar = 1 mm.

Lineage F comprised 10 unique haplotypes from 32 individuals (Figure 4a). There was a geographic structure in the haplotype network, as most sequences from the Pannonian basin were separated from Italian specimens by 10 mutational steps. However, there was also one shared haplotype between the two regions. Lineage G comprised 22 unique haplotypes from 78 individuals (Figure 4b). We identified a shallow geographic structure in the haplotype network, as most sequences from the northern coasts were separated from

Pannonic specimens by few mutational steps. Again, we observed shared haplotypes across the regions. All specimens from Great Britain belonged to one unique halotype with close similarity to haplotypes from the North Sea and Baltic Sea. The most distinct haplotypes originated from the Caspian Sea. In lineage H (with a total of 59 specimens), more than half of the sequences belonged to a single central haplotype, comprising individuals from northern Germany to Hungary (Figure 4c). A sequence from Armenia differed from the central haplotype by just one mutation. More distant haplotypes were obtained from the Balkan Peninsula and the surroundings of Vienna.

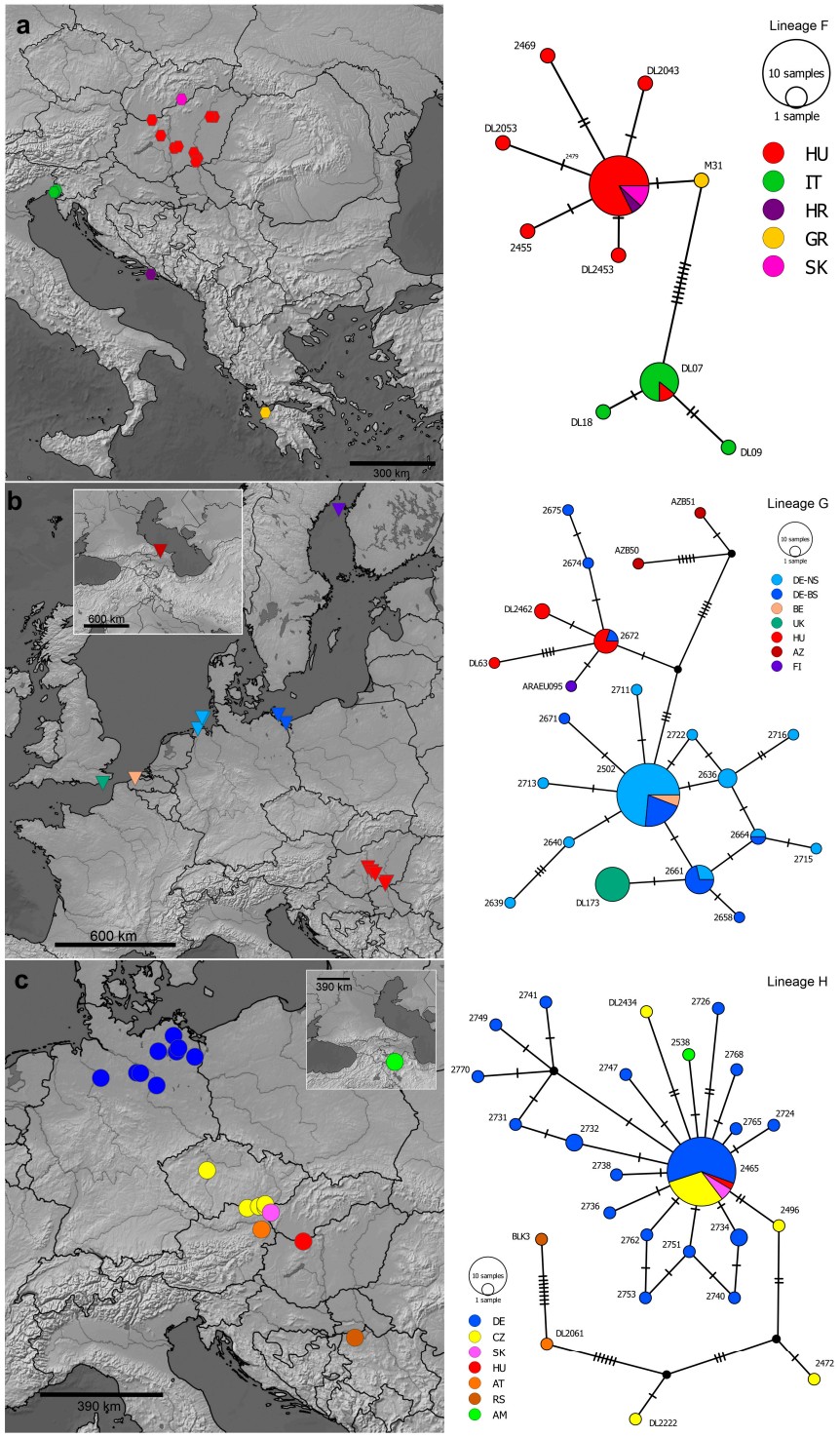

**Figure 4.** Haplotype networks and sampling locations of *Dactylochelifer latreillii* morphotype. (**a**) Lineage F; (**b**) lineage G; (**c**) lineage H. DE-NS = Germany, North Sea, DE-BS = Germany, Baltic Sea.

*3.3. Morphological Variation*

3.3.1. Multivariate Ratio Analysis

The results of shape PCR clearly demonstrate the morphological crypsis in *Dactylochelifer* pseudoscorpions. The three mtDNA lineages largely overlapped in morphospace, as they did in size (Figure 5). Lineage G showed the largest variation in size, while on average lineage H was the smallest. We observed no patterns of strong correlation between shape and size, and thus there was no indication for pronounced allometric effects.

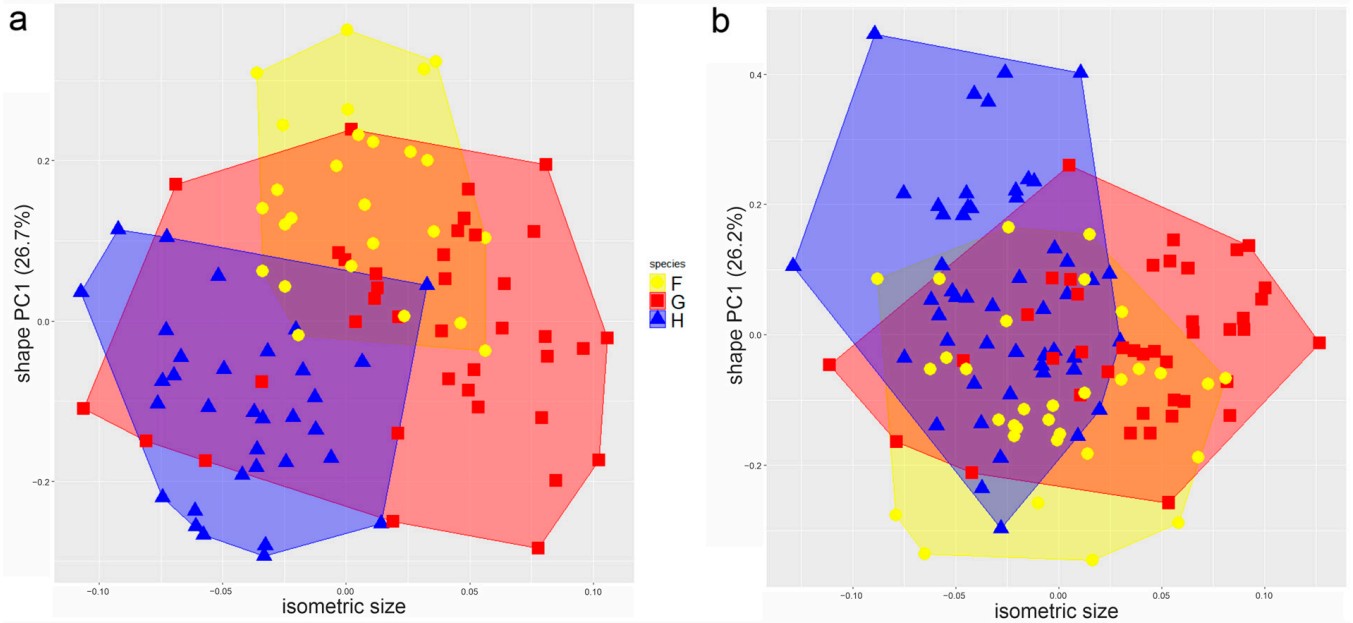

**Figure 5.** Scatterplot of isometric size against the first axis of shape PCA of the three *Dactylochelifer* lineages as identified by mtDNA. (**a**) Males; (**b**) females.

The high level of morphological crypsis was also reflected in the delimitation based on the LDA ratio extractor. Except for males of F and G, we found no morphometric ratios that would allow 100% separation of the lineages (Figure 6). On the other hand, the results clearly showed that morphological variation was far from being random. By including the best two ratios, a complete separation could be achieved for discrimination between males of lineages F and G (no overlap), and a good separation for males of lineages G and H (overlap 4.7%) and females of lineages F and H (overlap 10.3%). Discrimination was more challenging for females of lineages F and G (overlap 14.2%) and for males of lineages F and H (overlap 27.6%). Most difficult to separate were females of lineages G and H, and here the overlap was 52%. Fortunately, females of these lineages are clearly distinguishable by their genitalia (see below). Note that the overlap does not indicate the percentage of wrongly assigned specimens, it is simply the percentage of specimens that cannot be distinguished by these measurements. Separation of the lineages F-G and F-H was mainly due to shape rather than size ($\delta$ = 0.11–0.22, Table S4), whereas between lineages G-H a larger amount of the total separation was due to size ($\delta$ = 0.3–0.4, Table S4).

With few exceptions, the best discriminating ratios did not include proportions that were used traditionally in *Dactylochelifer* taxonomy, e.g., proportions of the palpal femur or male tarsus of leg I, or the relative length of the palpi. The variation in some of these proportions is shown in Figure 7. Specimens of lineage F (both sexes) can be tentatively differentiated by a more slender palpal femur, while in males of lineage H the male foretarsus is stouter on average, but there is overlap in all ratios.

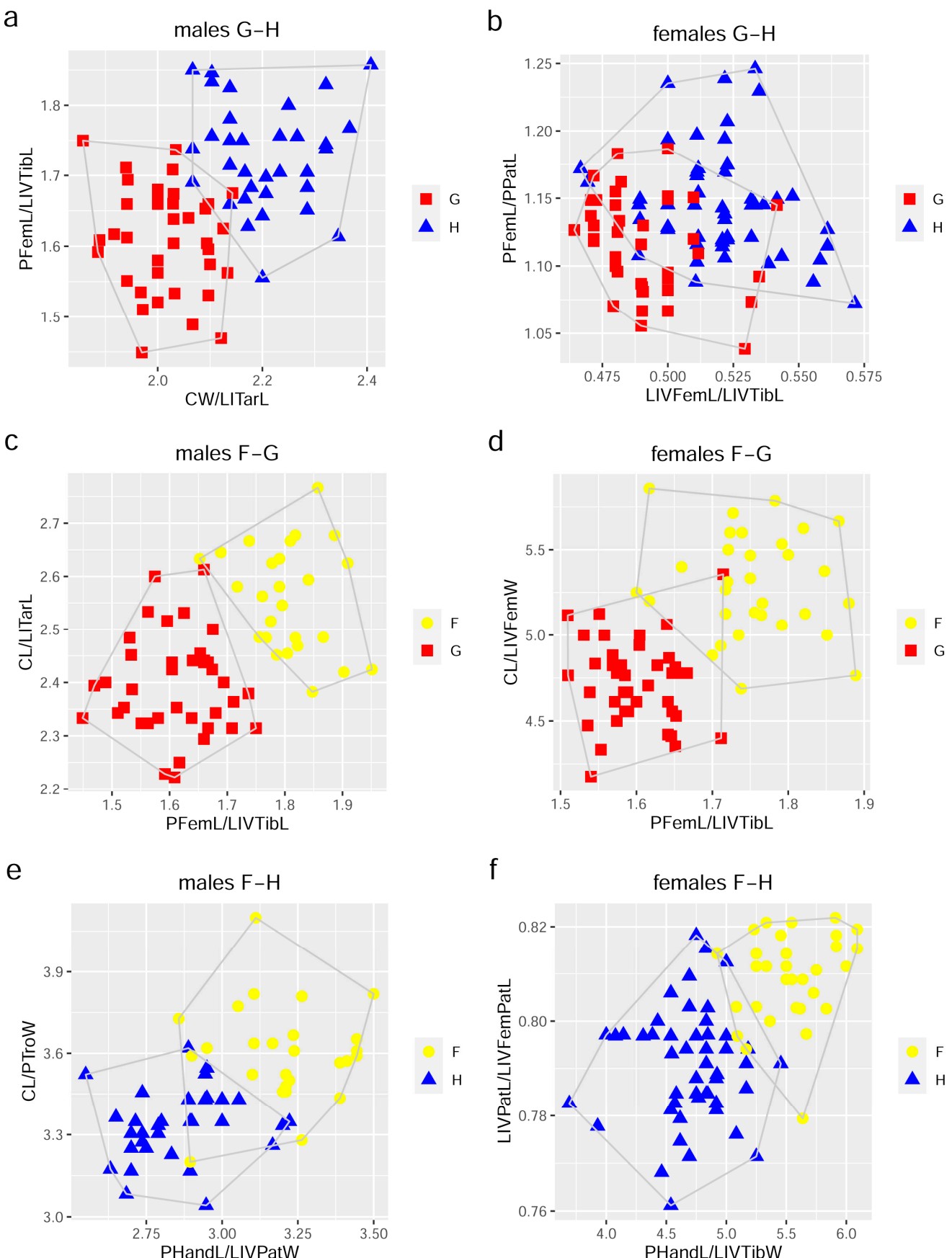

**Figure 6.** Scatterplots of the first versus the second most discriminating ratios in pairwise comparisons of the mtDNA lineages. (**a**,**c**,**e**) Males; (**b**,**d**,**f**) females. For measurement abbreviations, see Figure S1.

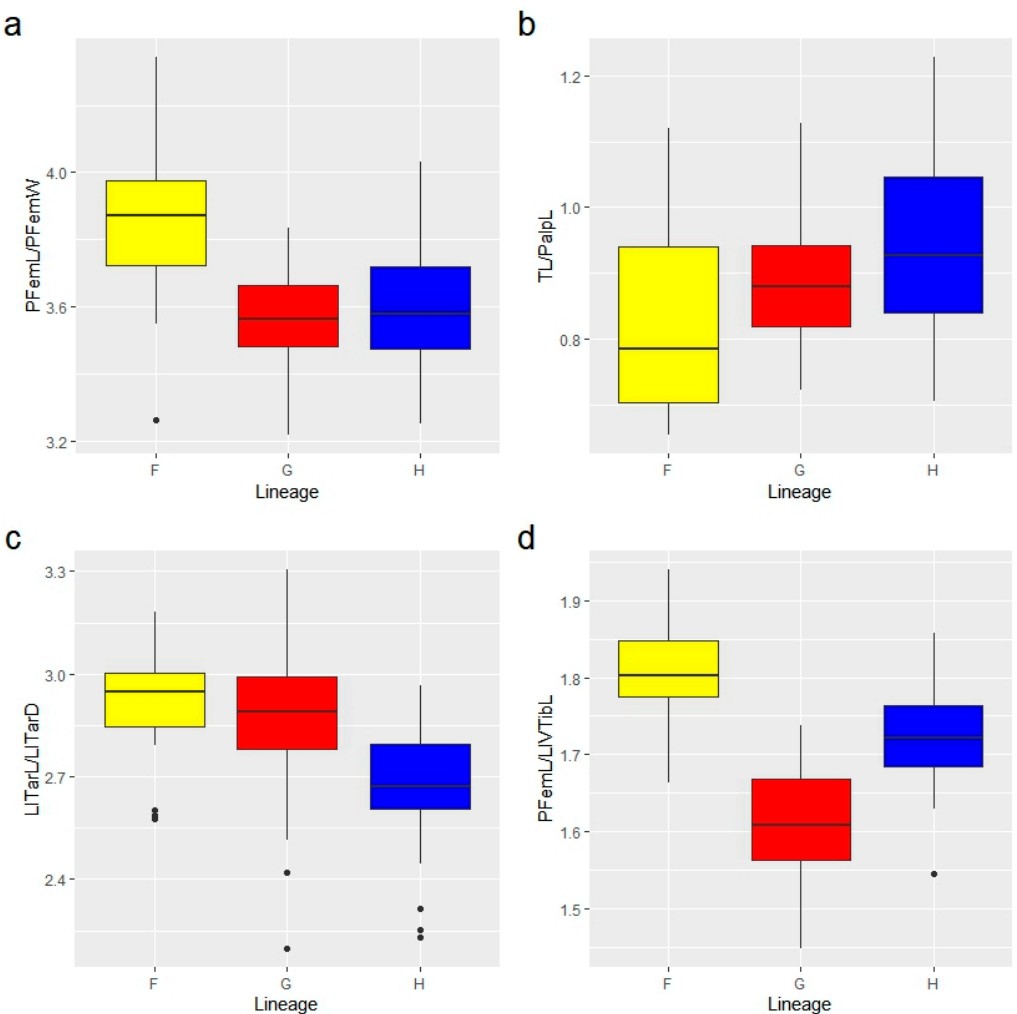

**Figure 7.** Boxplots of ratios that have been used traditionally to distinguish subspecies of *Dactyloche-lifer latreillii* (**a**–**c**) and one derived from multivariate ratio analysis (**d**). (**a**) Proportion of palpal femur; (**b**) length of pedipalp in relation to body size; (**c**) length/depth ratio of male tarsus I; (**d**) length ratio of palpal femur/tibia IV in males.

### 3.3.2. Shape Variation in Male Tarsus I

GM analysis revealed a substantial overlap in the shape of the male foretarsus, i.e., individual specimens could not be allocated with confidence to a certain lineage based on this character (Figure 8). PC1 accounted for 48.42% of variance while PC2 accounted for 22.38% of variance. The variation among PC1 was associated with the length/depth ratio of the tarsus, while the variation among PC2 was more associated with the shape and position of the dorsal mound/excavation (as shown on the thin plate splines). However, the mean shape of the male foretarsus differed significantly between the three lineages in all pairwise comparisons (ANOVA, $p < 0.05$).

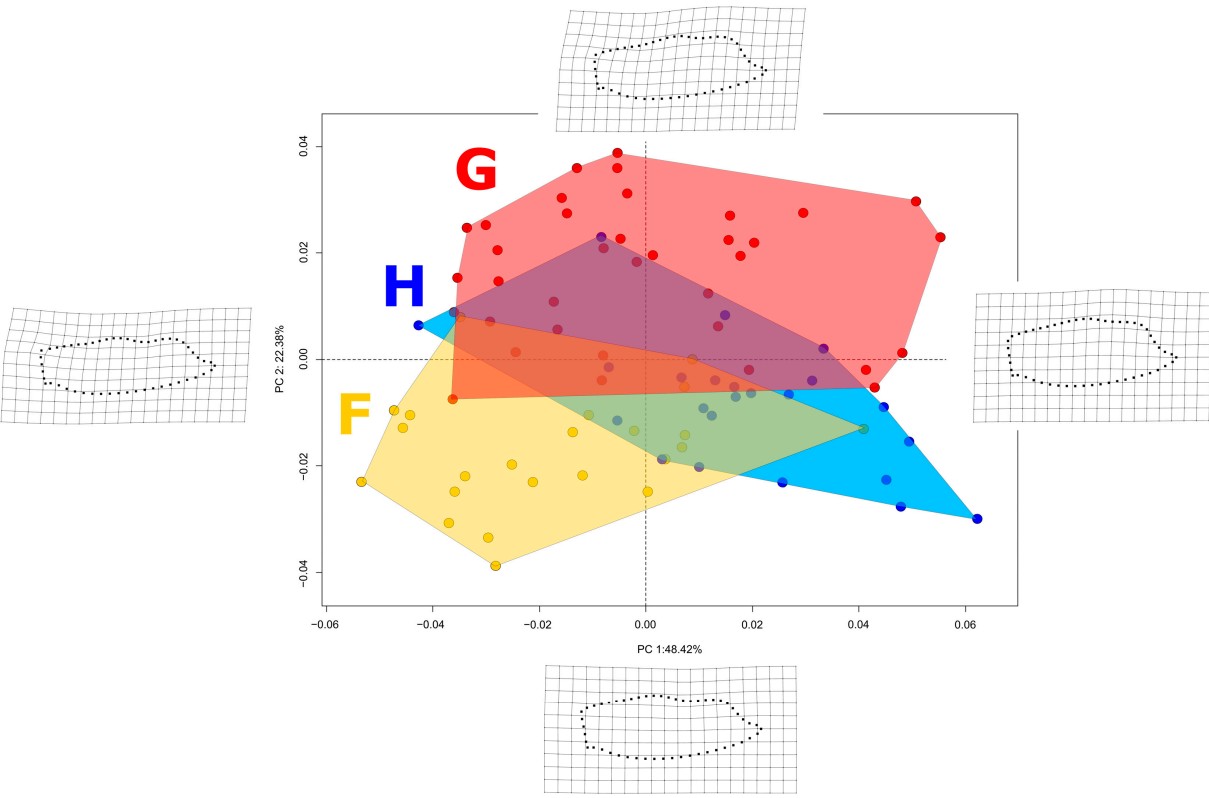

**Figure 8.** Shape analysis of the male tarsus of leg I in three lineages of the *Dactylochelifer latreillii* species complex. Shapes of the extremes along the PCR axes 1 and 2 are shown as thin plate splines.

### 3.3.3. Genitalic Morphology

The sclerotized cribriform plates of the internal female genitalia show discrete and constant differences among the three lineages. In lineage F (Figure 9a–c), the cribriform plates are fused by a broad stalk, in total shape resembling a bow tie. In lineage G (Figure 9d–f), the cribriform plates are connected by a small and heavily sclerotized stalk. The lateral structures are funnel-like, and the overall shape resembles a traditional telephone receiver. Lineage H (Figure 9g–i) shows paired cribriform plates, separated from each other by 1.2–1.5 times their diameter, and not connected by sclerotized structures but showing a small central arc. The exact positioning is very important, as slight differences in view strongly affect the perceived shape [62]. We could not distinguish median and lateral cribriform plates, and thus homologies with structures in other Cheliferidae remain unspecified.

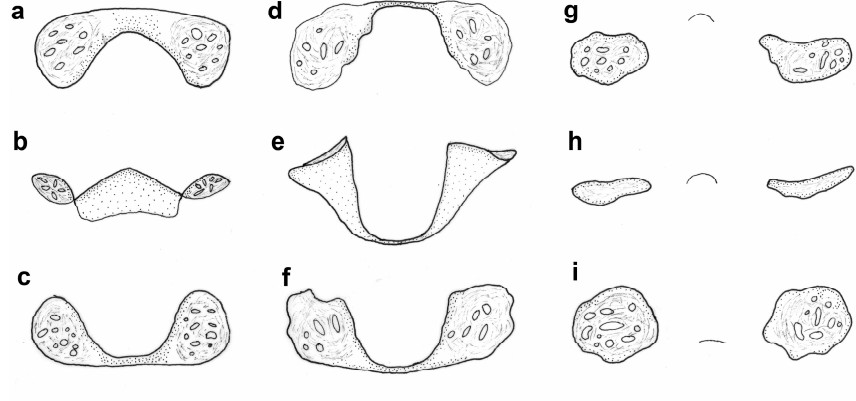

**Figure 9.** Structure of the cribriform plates in female genitalia. (**a**–**c**) Lineage F (*Dactylochelifer ninnii*); (**d**–**f**) Lineage G (*D. latreillii*); (**g**–**i**) Lineage H (*D. degeerii*). (**a**,**d**,**g**) Frontal view; (**b**,**e**,**h**) dorsal view; (**c**,**f**,**i**) caudal view. Scale 0.1 mm.

The structures of the internal male genitalia corresponded very well to the detailed descriptions by Vachon [37]. We could not identify differences among the three lineages.

*3.4. Taxonomy*

Thorough taxonomic investigation let us conclude that scientific names are available for all three lineages: lineage F = *Chelifer ninnii* Canestrini, 1876, lineage G = *Chelifer latreillii* Leach, 1817, and lineage H = *Chelifer degeerii* C.L. Koch, 1835. Here, we redefine and redescribe these species.

**Cheliferidae Risso, 1827**
**Dactylocheliferini Beier, 1932**
*Dactylochelifer* **Beier, 1932**

Within Dactylocheliferini (coxae of leg IV in males with well-differentiated atrium at the mouth of coxal sacs, male genitalia without sclerotic rod and anterior invagination at the statumen convolutum, cribriform plates in female genitalia fused into one medial plate), the genus *Dactylochelifer* is characterized by a movable finger of chelicera with one galeal seta, tarsal claws asymmetric and simple, subterminal setae of the tarsi simple, and tarsus of leg IV without a tactile seta [45,63]. *Dactylochelifer* is a diverse genus with 46 species and 6 subspecies currently accepted, mostly from the Palearctic [26]. Phylogenetic relationships within *Dactylochelifer* are largely unknown. Several groupings have been proposed based on the morphology of the tarsus of leg I in the males and/or the cribriform plates in females [64–66], but all authors considered these groups to be tentative. Here, we define the *Dactylochelifer latreillii* species complex in accordance with the delineation of *Dactylochelifer laterillii* in the current version of the World Pseudoscorpiones Catalog [26].

*Dactylochelifer latreillii* **species complex**

**Main characteristics.** Carapace longer than wide (up to 1.3×), with a pair of well-developed eyes, distinctly granulate, with two transverse furrows, anterior furrow deeper, slightly anterior to middle of carapace, posterior furrow in posterior fifth of carapace, with a total of 60–90 semiclavate setae. Tergites with scale-shaped microsculpture, divided by median suture line, without lateral modifications, tergites I–XI carrying 9–21 setae, tergite XI with an additional pair of long tactile setae. Sternites IV–XI with 8–18 setae, XI with an additional pair of long tactile setae. Fixed finger of chelicerae with 5 setae, moveable finger with one galeal seta, galea with up to six small apical rami, flagellum with three blades, the longest serrated in apical half. Pedipalp segments with distinct granulation, except for chelal fingers, trichobothriotaxy as figured by Gabbutt [38] (Figure 5) and Mahnert [67] (Figure 1), with trichobothrium *ist* standing slightly distal to *est*. Palpal femur 3.2–4.3× as long as wide. Leg segments without tactile setae. Male tarsus I moderately modified, more compact than in females, 2.2–3.3× as long as wide, with slight to moderate dorsal excavation in distal half (Figure S6). Male genitalia as figured by Vachon [37] (figure 17) with elongated statumen convolutum (4.5× longer than wide), apodemes of lateral genital sacs situated in the middle of statumen convolutum, and lateral genital sacs not reaching anterior statumen convolutum (we could not identify any differences among the three species treated below).

*Dactylochelifer latreillii* **(Leach, 1817)** (Figure S3)
*Chelifer latreillii* Leach, 1817 (Holotype male from Britain, BMNH, dry collection of Leach "135a", not examined).
*Dactylochelifer latreillii septentrionalis* Beier, 1932 **syn. nov.** (Holotype male from Nordfriesische Insel: Norderney, September 1911, R. Heymans leg., ZMHB 31926, examined).

**Diagnostic characters.** Characterized by the following unique COI substitutions (with reference to Alignment S1): 3-A, 72-A, 90-C, 136-C, 262-C, 264-T, 267-C, 297-C, 372-T, 393-T, 405-G, 534-C, 535-C. Cribriform plates in female genitalia fused to a single structure in the shape of a traditional telephone receiver (Figure 9d–f).

**Measurements.** *Males* (*n* = 39): Total length 2.64 (2.05–2.9). Carapace (length/width/ratio): 0.79 (0.69–0.84)/0.66 (0.59–0.71)/1.19 (1.13–1.26). Pedipalp (length/width/ratio): trochanter 0.39 (0.32–0.44)/0.23 (0.19–0.26)/1.68 (1.53–2.01), femur 0.76 (0.66–0.84)/0.21 (0.18–0.23)/3.59 (3.26–3.83), patella 0.68 (0.56–0.75)/0.26 (0.2–0.28)/2.68 (2.43–3.03), femur/patella length 1.12 (1.05–1.2), chela 1.22 (1.02–1.31)/0.34 (0.29–0.38)/3.59 (3.34–3.92), hand 0.59 (0.5–0.65)/0.34 (0.29–0.38)/1.74 (1.57–1.9), finger length 0.61 (0.5–0.64), hand/finger length 0.97 (0.88–1.06). Legs (length/depth/ratio): Leg I: tibia 0.28 (0.24–0.3)/0.13 (0.11–0.15)/2.23 (1.9–2.5), tarsus 0.33 (0.28–0.36)/0.12 (0.1–0.14)/2.86 (2.2–3.3); Leg IV tibia 0.47 (0.38–0.5)/0.13 (0.11–0.16)/3.74 (2.98–4.26), tarsus 0.39 (0.33–0.43)/0.089 (0.08–0.11)/4.45 (3.68–5.06). Pedipalp femur/leg IV tibia length 1.61 (1.45–1.74); carapace width/leg I tarsus length 2.02 (1.88–2.17); carapace length/leg IV tibia length 1.67 (1.55–1.82).

*Females* (*n* = 41): Total length 2.93 (2.2–3.65). Carapace (length/width/ratio): 0.82 (0.71–0.88)/0.71 (0.62–0.79)/1.15 (1.08–1.25). Pedipalp (length/width/ratio): trochanter 0.39 (0.34–0.44)/0.24 (0.2–0.27)/1.62 (1.42–1.78), femur 0.8 (0.69–0.89)/0.23 (0.2–0.25)/3.53 (3.22–3.79), patella 0.72 (0.63–0.79)/0.27 (0.23–0.3)/2.72 (2.48–2.92), femur/patella length 1.12 (1.04–1.19), chela 1.31 (1.13–1.4)/0.38 (0.33–0.42)/3.43 (3.08–3.73), hand 0.65 (0.55–0.72)/0.38 (0.33–0.42)/1.71 (1.57–1.95), finger length 0.63 (0.54–0.7), hand/finger length 1.03 (0.88–1.14). Legs (length/depth/ratio): Leg I: tibia 0.31 (0.25–0.34)/0.1 (0.09–0.12)/3.00 (2.52–3.29), tarsus 0.34 (0.29–0.37)/0.079 (0.07–0.1)/4.33 (3.45–5.0); Leg IV tibia 0.5 (0.42–0.56)/0.13 (0.11–0.14)/3.90 (3.07–4.29), tarsus 0.41 (0.36–0.45)/0.09 (0.08–0.1)/4.46 (3.81–4.92). Pedipalp femur/leg IV tibia length 1.60 (1.51–1.73); carapace width/leg I tarsus length 2.12 (1.86–2.40); carapace length/leg IV tibia length 1.63 (1.41–1.83).

**Color.** Carapace, dark brown with posterior disk conspicuously lighter. Palps, dark brown, distinctly reddish towards the joints, hand almost black, fingers reddish (Figures 2a and 3). Legs yellowish to reddish brown. Abdominal tergites light brown, divided by fine yellowish midline, each half-tergite with dark brown spot except for tergites I, III, XI. Sternites yellowish brown with indistinct dark spots at half-sternites IV–X.

**Habitat.** On the British Islands and in northern Europe it is a strictly maritime species [68–70], most abundant in marram grass (*Ammophila*) in sand dunes, sometimes under pieces of wood, in strand-line debris, and under bark at localities close to the sea; in Scotland and Scandinavia on rocks of the splash zone often associated with *Parmelia* lichens [71]. In Sweden, the species is named "strandklokrypare" [72]. In Hungary, the habitat remains unspecified, but distribution of the localities corresponds strongly with areas of salt-affected soils and inland salt habitats [73].

**Distribution.** Verified records from the Atlantic coast from the British Islands and France to Denmark, around the Baltic Sea, and from the Caspian Sea. Isolated(?) records from Hungary. Currently no evidence for occurrence in the Mediterranean.

**Remarks on synonymy.** The correspondence of the type specimen of *Chelifer latreillii* Leach, 1817 with the species currently recognized as *Dactylochelifer latreillii* in Great Britain has repeatedly been confirmed [74–76]. Since the species is strictly maritime in Great Britain, and from coastal habitats of central and northern Europe only a single DNA lineage (G) has been recorded, which includes British specimens, the synonymy of *Dactylochelifer latreillii septentrionalis* Beier, 1932, described from the German North Sea Island Norderney, seems well substantiated. Note: in the World Pseudoscorpiones Catalog [26] the type locality of *D. l. septentrionalis* is erroneously cited as "Friesland, The Netherlands".

*Dactylochelifer degeerii* **(C.L. Koch, 1835)** (Figure S4)
*Chelifer degeerii* C.L. Koch, 1835 (type material from Regensburg, Bavaria, Germany, probably lost).
*Chelifer fabricii* C.L. Koch, 1835 (type material from Regensburg, Bavaria, Germany, probably lost).
*Chelifer angustus* C.L. Koch, 1836 (type material from Regensburg, Bavaria, Germany, probably lost).
*Chelifer schaefferi* C.L. Koch, 1839 (type material from Bavaria, Germany, probably lost).

*Chelifer brevipalpis* Canestrini, 1876 (type material from Bosco di Cervarese, Padova, Veneto, Italy, probably lost [77].

**Diagnostic characters.** Characterized by the following unique COI substitutions (with reference to Alignment S1): 90-A, 165-C, 222-C, 258-G, 267-A, 297-T, 333-T, 372-C, 429-G, 459-T, 468-C, 487-A, 498-A, 513-G. Cribriform plates in female genitalia not connected, separated approximately by 1.2–1.5 times of their diameter (Figure 9g–i).

**Measurements.** *Males* (*n* = 33): Total length 2.38 (2.05–2.75). Carapace (length/width/ratio): 0.75 (0.7–0.81)/0.64 (0.61–0.71)/1.17 (1.1–1.25). Pedipalp (length/width/ratio): trochanter 0.35 (0.32–0.39)/0.23 (0.21–0.24)/1.56 (1.36–1.76), femur 0.73 (0.69–0.84)/0.2 (0.17–0.22)/3.68 (3.33–4.01), patella 0.63 (0.58–0.74)/0.24 (0.21–0.27)/2.62 (2.3–2.86), femur/patella length 1.16 (1.05–1.26), chela 1.13 (1.08–1.25)/0.32 (0.28–0.35)/3.54 (3.31–3.82), hand 0.54 (0.5–0.64)/0.32 (0.28–0.35)/1.70 (1.57–1.91), finger length 0.56 (0.52–0.6), hand/finger length 0.97 (0.88–1.1). Legs (length/depth/ratio): Leg I: tibia 0.25 (0.23–0.29)/0.11 (0.1–0.13)/2.22 (1.81–2.5), tarsus 0.29 (0.27–0.31)/0.11 (0.1–0.13)/2.66 (2.23–2.97); Leg IV tibia 0.42 (0.39–0.48)/0.12 (0.1–0.15)/3.59 (2.73–4.09), tarsus 0.36 (0.34–0.4)/0.085 (0.08–0.09)/4.35 (3.8–4.69). Pedipalp femur/leg IV tibia length 1.73 (1.55–1.86); carapace width/leg I tarsus length 2.20 (2.05–2.44); carapace length/leg IV tibia length 1.78 (1.67–1.89).

*Females* (*n* = 51): Total length 3.02 (2.37–3.72). Carapace (length/width/ratio): 0.77 (0.6–0.84)/0.68 (0.54–0.75)/1.13 (1.01–1.22). Pedipalp (length/width/ratio): trochanter 0.36 (0.31–0.4)/0.23 (0.19–0.26)/1.57 (1.36–1.82), femur 0.75 (0.62–0.84)/0.21 (0.18–0.23)/3.55 (3.25–4.03), patella 0.66 (0.56–0.7)/0.26 (0.21–0.28)/2.57 (2.19–2.83), femur/patella length 1.14 (1.06–1.24), chela 1.20 (1.02–1.29)/0.36 (0.3–0.39)/3.32(3.1–3.63), hand 0.59 (0.48–0.64)/0.36 (0.3–0.39)/1.63 (1.49–1.81), finger length 0.59 (0.5–0.65), hand/finger length 0.99 (0.86–1.08). Legs (length/depth/ratio): Leg I: tibia 0.29 (0.25–0.31)/0.099 (0.09–0.11)/2.88 (2.58–3.12), tarsus 0.31 (0.27–0.33)/0.074 (0.07–0.09)/4.19 (3.72–4.55); Leg IV tibia 0.45 (0.39–0.49)/0.13 (0.11–0.15)/3.6 (2.83–4.07), tarsus 0.38 (0.33–0.42)/0.09 (0.08–0.11)/4.30 (3.82–4.73). Pedipalp femur/leg IV tibia length 1.67 (1.35–1.80); carapace width/leg I tarsus length 2.2 (1.62–2.58); carapace length/leg IV tibia length 1.71 (1.31–1.87).

**Color.** Carapace, dark brown with posterior disk merely slightly lighter than median disk. Palps, dark brown, distinctly reddish towards the joints, hand almost black, fingers reddish (Figure 3). Legs yellowish to light brown. Abdominal tergites brown, divided by fine yellowish midline, each half-tergite with dark brown spot except for tergites I, III, XI. Sternites light brown with indistinct dark spots at half-sternites IV–X.

**Habitat.** This is an inland species, primarily associated with floodplains of rivers and rivulets [78,79]. Most literature records refer to specimens from bark and dead wood [80–82], but preferred microhabitat may actually be bushes, as already pointed out by C. L. Koch [83] "Im Sommer und Frühjahre bewohnt er gern niederes Gebüsch und wird dann zuweilen von diesem herunter geklopft." Frequently found in bird nests [84,85] and rarely in compost heaps [80,86].

**Distribution.** Verified records from inland habitats in Germany, the Benelux countries, Switzerland, Austria, the Czech Republic, Slovakia, Hungary, Serbia, Italy and Armenia. Probably widespread in the continental biogeographic region of Europe. The distribution in the Mediterranean needs to be studied in detail. So far, we assigned specimens from northern Italy to this species.

**Remarks on synonymy.** This species corresponds to *Dactylochelifer latreillii latreillii* in the sense of Beier [45,79] and most subsequent authors. In inland habitats of central Europe north of the Alps, only one mitochondrial lineage (H) occurs. Taking into account recent dating of C.L. Koch´s publications [87], the oldest available name for specimens from that region is *Chelifer degeerii* C.L. Koch, described from Regensburg in Bavaria [88]. Although some material of species described by C.L. Koch is housed at BMNH in London, this material originates from Nuremberg, where L. Koch—the son of C.L. Koch—was active, and it was not considered as a type material for species described by C.L. Koch [76]. The descriptions and illustrations of *C. degeeri* in C.L. Koch's books [83,88] correspond perfectly to recently collected material from the mitochondrial H lineage. C.L. Koch further described

three species from the region of Regensburg [89,90] that are considered junior synonyms of one and the same species. In light of the homogeneity of the mitochondrial H-lineage, this view is corroborated by our study.

We examined material from several inland localities in Italy from the Gardini collection (Table S2) that we allocated to *D. degeerii*, based on the unique structure of the median cribriform plate, the best discriminating morphometric ratios, and coloration (we failed to generate COI sequences from this old material). Therefore, we propose that Canestrinis *Chelifer brevipalpis*, described from an inland forest near Padova [91], is a junior synonym of *D. degeerii*. The shortness of the palpi was a main diagnostic character in the original description, and on average *D. degeeri* has, indeed, the shortest palpi among the three species (Figure 7b), although overlap is substantially. For further differences from *D. ninnii* see remarks on this species.

*Dactylochelifer ninnii* **(Canestrini, 1876)** (Figure S5)
*Chelifer ninnii* Canestrini, 1876 (type material from Valle Dogado, Laguna di Venezia, Italy, present in Canestrini collection at Zoology Museum of Padova University [77], image examined).

**Diagnostic characters.** Characterized by the following unique COI substitutions (with reference to Alignment S1): 15-C, 54-T, 90-T, 156-G, 162-G, 231-T, 267-T, 270-G, 285-A, 297-A, 306-G, 315-T, 342-A, 354-C, 378-C, 396-C, 402-C, 420-G, 426-G, 465-A, 483-T, 486-G, 496-C, 543-A. Cribriform plates in female genitalia fused to a single structure in the shape of a bow tie (Figure 9a–c). Color little-contrasting, palpal segments weakly bicolored, integument of carapace and palps conspicuously shining. Palpal femur more slender (mean length/width ratio 3.85). Fingers of chela bent, longer than hand with pedicel.

**Measurements.** *Males* (*n* = 25): Total length 2.38 (2.05–2.9). Carapace (length/width/ratio): 0.81 (0.75–0.84)/0.66 (0.63–0.7)/1.21 (1.16–1.27). Pedipalp (length/width/ratio): trochanter 0.37 (0.33–0.41)/0.23 (0.2–0.25)/1.67 (1.46–1.84), femur 0.80 (0.73–0.87)/0.21 (0.18–0.23)/3.85 (3.26–4.34), patella 0.69 (0.63–0.75)/0.25 (0.21–0.26)/2.80 (2.54–3.05), femur/patella length 1.15 (1.09–1.25), chela 1.22 (1.15–1.32)/0.33 (0.27–0.38)/3.73 (3.17–4.48), hand 0.60 (0.55–0.68)/0.33 (0.27–0.38)/1.84 (1.48–2.15), finger length 0.59 (0.54–0.63), hand/finger length 1.02 (0.9–1.11). Legs (length/depth/ratio): Leg I: tibia 0.27 (0.26–0.29)/0.12 (0.11–0.14)/2.31 (1.98–2.6), tarsus 0.32 (0.27–0.34)/0.11 (0.1–0.12)/2.92 (2.58–3.18); Leg IV tibia 0.44 (0.41–0.49)/0.12 (0.1–0.13)/3.83 (3.2–4.23), tarsus 0.38 (0.36–0.41)/0.086 (0.08–0.09)/4.50 (4.06–5.04). Pedipalp femur/leg IV tibia length 1.81 (1.66–1.94); carapace width/leg I tarsus length 2.12 (1.92–2.36); carapace length/leg IV tibia length 1.83 (1.69–1.96).

*Females* (*n* = 32): Total length 2.81 (2.16–3.74). Carapace (length/width/ratio): 0.83 (0.75–0.93)/0.7 (0.65–0.76)/1.19 (1.1–1.27). Pedipalp (length/width/ratio): trochanter 0.38 (0.32–0.45)/0.23 (0.21–0.25)/1.63 (1.42–1.81), femur 0.82 (0.73–0.94)/0.21 (0.18–0.25)/3.86 (3.55–4.19), patella 0.71 (0.62–0.79)/0.25 (0.23–0.28)/2.80 (2.6–3.03), femur/patella length 1.16 (1.08–1.23), chela 1.29 (1.17–1.43)/0.36 (0.31–0.42)/3.63 (3.39–4.04), hand 0.65 (0.56–0.73)/0.36 (0.31–0.42)/1.82 (1.68–2.01), finger length 0.62 (0.57–0.67), hand/finger length 1.05 (0.98–1.12). Legs (length/depth/ratio): Leg I: tibia 0.29 (0.26–0.32)/0.098 (0.09–0.11)/2.96 (2.75–3.4), tarsus 0.33 (0.28–0.35)/0.077 (0.07–0.09)/4.33 (3.55–4.97); Leg IV tibia 0.47 (0.42–0.51)/0.12 (0.11–0.13)/4.01 (3.69–4.4), tarsus 0.4 (0.36–0.43)/0.09 (0.08–0.1)/4.53 (3.91–5.0). Pedipalp femur/leg IV tibia length 1.75 (1.61–1.89); carapace width/leg I tarsus length 2.13 (1.96–2.38); carapace length/leg IV tibia length 1.76 (1.65–1.88).

**Color.** Carapace and palps uniformly colored in greyish brown (Figure 3), posterior disk and palpal fingers lighter, all segments conspicuously shining. Abdomen dorsally pale, with the dark spots at tergites II and IV-X turning up contrasting, venter darker than dorsum. Legs yellowish to light brown.

**Habitat.** Canestrini described this species from specimens that were found under the bark of poles that stood for years in the salt water of the Venice lagoon [91]. As our sequenced material originated either from the Mediterranean coast or from Pannonic salt steppes, an association with salt-affected habitats is likely. However, because most

specimens in this study originate from reed galls, the original habitat cannot be inferred in most cases.

**Distribution.** Verified records from Italy (Adriatic, Ionian, Tyrrhenian coast, Sardegna), Croatia, Greece (Ionian coast), France (Côte d'Azur), and from the Pannonian Basin (Hungary, Slovakia). To our current knowledge this is a southern species, with the northern distribution limits corresponding to the northernmost outposts of Pannonic salt steppes [92]. This species may be widespread in the Mediterranean, particularly in the central Mediterranean, where congeners with either stout, strongly modified male tarsus I or slender, not clearly modified foretarsus do not occur [64].

**Remarks on synonymy.** The distinctiveness of a Mediterranean form of *Chelifer latreillii* was recognized long ago [41,42]. We examined and sequenced a large series of specimens from Villaggio del Pescatore (Friuli-Venezia Giulia) on the Adriatic coast, which is only some 100 km from the type locality of *C. ninnii*. All specimens belonged to a single lineage and they fitted perfectly to the description by Canestrini [91]. In particular, they differed from *C. brevipalpis* in the relative length of the palps (longer than body), the shape of the palpal trochanter, the hand/finger ratio (fingers longer than hand) and the brighter color. We could not examine the type material, but by courtesy of Luis Guariento we have received access to handwritten notices of Canestrini dating back to 1881–1883 ("Note di Giovanni Canestrini 1881, 1882, 1883" stored in the Biblioteca del Museo Tridentino di Scienze Naturali, Trento, n. 45036). From these we could conclude that the marvelous color illustration of *C. degeeri* by Canestrini [93] refers to *C. ninnii*, that definitely matches our specimens, while a color illustration of *C. brevipalpis* was retained unpublished, probably in consequence of the synonymization by Simon [39]. Also, images of the type specimen, kindly provided by Luis Guariento, support our conclusion.

Without examination of the type material we felt unable to decide on the identity of *Chelifer pediculoides* Lucas, 1849, which may be a senior synonym of *C. ninnii*. This species was described from Oran, Algeria, and the type may be preserved at MNHN Paris [40]. Our attempts to gain access to the type material failed. The description of Lucas [94] generally fits to our specimens, but Lucas did not refer to the shape and size of the male leg tarsus I. In consideration of the many *Dactylochelifer* species that have been described since Simon's inspection of the type of *pediculoides* [40] from the western Mediterranean [62,64,95,96], mostly diagnosed by characters of the male foretarsus, we remain cautious in suggesting synonymy.

The range of lineage F comprises the terra typica of *Chelifer cephalonicus* Beier, 1929, later considered a subspecies of *Dactylochelifer latreillii* by Beier [45,79]. Mahnert [46] synonymizd *cephalonicus* with *latreillii* with reference to substantial variations in the proportions of the palpal femur and male leg tarsus I. However, in size and several proportions (palpal femur), the type of *C. cephalonicus* falls outside the range of all three species as delineated in this study, and the shape of the male foretarsus also deviates. We suspect that *cephalonicus* may be a distinct species, and Mahnert [46] may have included specimens of *C. ninnii* in his measurements. Callaini [97] provided a detailed description of two forms of *Dactylochelifer latreillii latreillii* from southern Italy. Forma β corresponds in all measurements and proportions to *D. ninnii* and it is very likely that Callaini characterized this species, whereas forma α probably matches *cephalonicus*. We examined specimens from several localities in Italy (Veneto: Bibione, Caorle, Pellestrina; Sicily; see Table S2) with conspicuous short palpal femora that fitted well to the description of *cephalonicus*. A trans-Ionian distribution of *cephalonicus* in Greece and Italy has already been proposed by Lazzeroni [98]. However, for the reinstatement of *D. cephalonicus* as a separate species, we will await confirmation by DNA data.

## 4. Discussion

Our study revealed a further complex of cryptic species in central European pseudoscorpions. Species delineation in the *Dactylochelifer latreillii* species complex is particularly challenging, because morphological stasis at the evolutionary timescale is faced

in conjunction with high levels of intraspecific variation, even within populations. Most importantly, we demonstrated substantial intraspecific variation in the shape of the tarsus of the first leg in males, which was generally considered the most reliable diagnostic characteristic in *Dactylochelifer* taxonomy. This view was advanced by Beier [99] when he wrote "Bei dieser Gelegenheit möchte ich nochmals auf die Form des männlichen Vordertarsus als auf ein systematisch wichtiges und äußerst brauchbares Merkmal hinweisen. Mit dessen Hilfe lassen sich auch Arten, die einander sehr nahestehen und nach der Form der Palpen kaum zu trennen sind, noch gut unterscheiden. Leider ist die Bedeutung dieses Merkmales bisher noch nicht genügend gewürdigt worden" [On this occasion I would like emphasise the shape of the male foretarsus as a systematically important and extremely useful feature. With its help, species that are very close to each other and can hardly be separated based on the shape of the palps can still be easily distinguished. Unfortunately, the importance of this feature has not yet been sufficiently appreciated]. Subsequently, many species have been described that were mainly or entirely based on this characteristic (summary in [64]). The tarsal morphology of leg I is believed to be of functional significance during sperm transfer [29]. *Dactylochelifer* males deposit spermatophores that are among the most complex ones in Pseudoscorpiones, and they are directly transferred following impressive mating dances [28,100]. In the final phase of the courtship ceremony, the male stretches his modified first pair of legs and opens the genital operculum of the female, while facing the exterior tarsal margin towards the stalk of the spermatophore, thereby bordering and stabilizing the structure [29]. Weygoldt hypothesized that the male foretarsus may press against the sperm package and force the swollen sperm mass into the seminal receptacle [101]. Our data suggest that this character may be less specific than previously assumed. We demonstrate large overlap in the proportion and shape of the male foretarsus across species boundaries (Figure 8, see also Figure S6). We also noticed remarkable variation within populations, almost equaling the range-wide variation within species. Overrating the diagnostic value of this trait may result in false conclusions in two respects: oversplitting by ignoring intraspecific variation, and otherwise overlooking hidden diversity that is not expressed in this character.

We conclude that the correct recognition of species boundaries in this species complex is almost impossible without the aid of DNA data. In contrast, the delineation of species entities using the DNA barcoding gene was straightforward in this case, as variation among lineages was more than 10 times higher than variation within lineages. But genetic divergence is not reflected in morphological distinctiveness, as is characteristic for cryptic species [102]. The three species widely overlap in morphospace, and we could not detect any single measurement or proportion that would allow unequivocal identification. Nevertheless, morphological variation is far from being random. This is shown in the scatterplots of the best discriminating ratios for pairwise comparisons as determined in the framework of multivariate ratio analysis (Figure 6). Although there is overlap, these graphs can be of practical use for the identification of many specimens (those that fall outside the overlapping zone) by just taking four measurements. Notably, only few ratios contain measurements that have been considered useful for species delineation before, demonstrating the explorative power of MRA in the analysis of cryptic species. At least for the practical determination of pseudoscorpions, we consider this approach superior to the propagation of discriminant functions that require the collection of multiple measurements and count data [15] that are hard to gather with standard laboratory facilities.

Further potential for species delimitation could be seen in genitalic variation. Genitalic characters are only infrequently considered in taxonomic studies of Pseudoscorpiones, primarily owing to the internal location that requires preparation and partial destruction of the specimens for examination [103]. Pseudoscorpion genitalia have been considered useful for diagnoses at higher taxonomic levels [103,104], but they have hardly ever been used for delineation at the species level [37,105,106]. For Cheliferidae, and for *Dactylochelifer* in particular, Vachon [37] proposed that the configuration of the median cribriform plate in female genitalia could be useful for species discrimination. Since then, the struc-

ture of the cribriform plates has been figured in descriptions of several *Dactylochelifer* species [62,65,66,107]. Mahnert [62] predicted limited usefulness for discrimination at the species level, but potential value for inferring relationships among species. Here, our results contradict. By not understanding *Dactylochelifer latreillii* as a species complex, Mahnert may have misjudged the diagnostic value for species identification by overrating variation within species. We found discrete differences in the shape of the cribriform plates among the species within the *Dactylochelifer latreillii* complex. In fact, these are the best morphological characteristics to separate the species. In particular, the configuration of the cribriform plates in *D. latreillii* (lineage H) is highly characteristic within the genus, as the plates are not fused but widely separated, resembling, for example, the arrangement in *Centrochelifer* [65] (figure 44). Interestingly, a paired, not fused median cribriform plate has been considered a diagnostic characteristic of the tribe Cheliferini, while Dactylocheliferini should have a fused median cribriform plate [45,63]. Thus, our results rather question the systematic value at higher taxonomic levels. It should be noted that homologies among structures of the cribriform plates are not fully understood, but this would not reduce the value for species discrimination. In males, on the other hand, the genital aperture differed from more distinct *Dactylochelifer* species [65,66,108] but was not differentiable among the three investigated species of the *latreillii* complex.

Last but not least, the species are clearly separated ecologically, at least in large parts of their areas. In Europe, north of the Alps we face a pattern of strictly parapatric distribution with abutting but nonoverlapping ranges: *Dactylochelifer latreillii* is strictly coastal, whereas *D. degeerii* has never been found in close proximity to the sea. The situation is more complex in the Pannonian basin, where all three species occur sympatrically (the maintenance of genetic differentiation in sympatry can be taken as an additional argument for the status of the lineages as biospecies). Currently, we cannot infer niche differentiation in this region, as our material from Hungary originated principally from cigar-like galls on the common reed (*Phragmites australis*) which are induced by the fruit fly genus *Lipara*. The galls were selectively collected in the course of comprehensive projects [109,110], resulting in a strong sampling bias in our material. Probably, the galls are not the primary habitat for the pseudoscorpions, but these were encountered accidently during phoretic transport via flying insects [25]. Phoresy is known from *Dactylochelifer* species [66], and this is also suggested by the frequent presence in bird nests [81,84,85]. The detection of a shared haplotype occurring in Hungary and on the Baltic Sea coast (*Dactylochelifer latreillii*), and an almost identical haplotype occurring in central Europa and Armenia (*Dactylochelifer degeerii*), suggests the effective incidence of occasional long distance dispersal. But we do not know to what extent the inferred distribution patterns are affected by phoretic dispersal. Also, from the Mediterranean the available data on habitat and distribution are scarce and biased. Taking the overall picture, we hypothesize that the distribution in this species complex is mainly driven by species-specific salt tolerances. *Dactylochelifer latreillii* is a halobiont species, only recorded from coastal habitats and salt-affected inland sites in the Pannonian basin. Marram grass (*Ammophila arenaria*), being the preferred vegetation structure in northern latitudes, does not occur in Hungary, and there the species may thrive in tussocks of salt marsh plants such as *Puccinellia limosa* [73]. Halophilic tendencies can also be derived for *D. ninnii* from the known distribution records, which furthermore suggest higher thermal preference. *Dactylochelifer degeerii*, on the other hand, is an inland species that never has been recorded from salt-affected habitats.

This is the point at which to appreciate once more the contributions of the old authors, whose legacies are diverse and inconsistent, depending on individual skills and the respective facilities of their times and places. Carl Ludwig Koch (1778–1857) was ineffective in the description of several *Dactylochelifer* species from the surroundings of Regensburg that later turned out to be mere variants of one and the same species. However, he was the first to recognize that *Dactylochelifer degeerii* lives in the higher vegetation of shrubs, from where it can be beaten [83,88,89]. This knowledge has been forgotten for more than 100 years. According to contemporary catalogs and lists, *Dactylochelifer* spp. would be relatively rare in

central Europe and mainly collected from under bark and rotten wood [80,111,112]. In the face of the steady and abundant occurrence of *Dactylochelifer degeerii* in *Humulus*-overgrown shrubs in the north-eastern German plain, the records from tree-related microhabitats (as well as those from bird nests) appear rather accidental. The preferred habitat has simply not been sampled by pseudoscorpionologists, at least not with the appropriate technique (strong beating above a white flower pot coaster). Giovanni Canestrini (1835–1900), on the other hand, was a precise observer with a reliable instinct for species delimitation even in delicate situations. He correctly recognized the existence of two *Dactylochelifer* species in the surroundings of Padova [91], and the detailed descriptions and illustrations are still useful for identification. Ad partim, even the diagnostic characters are still of practical value. Unfortunately, Canestrini [93] later followed the authority of Simon [39] and considered both *brevipalpis* and *ninnii* as variants of the supposedly widespread *degeeri*, which caused a distinct *Dactylochelifer* species to be forgotten for more than a century. We strongly encourage contemporary researchers to study the historical literature seriously, because modern taxonomy can benefit a lot from the buried knowledge.

A continent-wide revision of *Dactylochelifer* was surely beyond the scope of this study, but it has to be noted that the *Dactylochelifer latreillii* species complex may include more species from the Mediterranean, Asia Minor, and the Near East. Many similar *Dactylochelifer* species were described from central Asia, and for some of them, e.g., *D. amurensis* (Tullgren, 1907) and *D. redikorzevi* (Beier, 1929), upcoming synonymy with *D. latreillii* was predicted [113]. A closely related species from the Mediterranean is *D. balearicus* Beier, 1961 from the Balearic Islands. The great similarity with *D. latreillii* was already highlighted in the original description [114]. We examined the syntype series from Menorca (three males, two females, NHMW 23220) and found it to be almost identical to *D. ninnii*, except for the somewhat shorter palpal fingers. But given the high degree of morphological crypsis in this genus, we will not propose nomenclatorial consequences until DNA barcodes corroborate the synonymy. Also, *D. pallidus* Beier, 1963 from Israel corresponds in all details of its original description with *D. ninnii*, except for the presence of teeth at the posterior claw of the male foretarsus. As already stated above, the species status of *D. cephalonicus* also needs to be reconsidered. There remains a lot of hidden diversity in the Mediterranean that awaits thorough study. Eventually, the distribution area of *Dactylochelifer latreillii* s. l., which—based on published records [26]—includes most countries of Europa, parts of central Asia (Kazakhstan), and the Middle East (Iran), and a stretch of the Mahgreb region (Algeria, Tunisia), will need to be redefined.

**Supplementary Materials:** The following supporting information can be downloaded at: https://www.mdpi.com/article/10.3390/d16030137/s1, Figure S1: Illustration of measurements; Figure S2: Geometric morphometry: landmarks and semi-landmarks; Figure S3: SEM images of *Dactylochelifer latreillii* (Leach, 1817) (lineage G); Figure S4: SEM images of *Dactylochelifer degeerii* (C. L. Koch, 1835) (lineage H); Figure S5: SEM images of *Dactylochelifer ninnii* (Canestrini, 1876) (lineage F); Figure S6: Variation in the shape of the male foretarsus in *Dactylochelifer* lineages/species; Table S1: Material freshly collected in the course of this study (2022–2023); Table S2: Examined material from museums (including types); Table S3: List of materials used for sequencing, morphometric, and GM analyses (including measurements and GenBank accession numbers); Table S4: Best discriminating ratios in pairwise comparisons from multivariate ratio analysis (MRA); Alignment S1: Alignment "Dactylochelifer_total" containing 170 COI sequences (including out-group).

**Author Contributions:** Conceptualization, C.M. and F.Š.; validation, F.Š. and C.M.; formal analysis, C.M., F.Š. and J.K.; investigation, F.Š., C.M., P.B. and P.H.; resources, F.Š., P.B., P.H. and C.M.; data curation, C.M. and F.Š.; writing—original draft preparation, C.M.; writing—review and editing, F.Š. and J.K.; visualization, F.Š., C.M. and J.K.; supervision, F.Š.; project administration, C.M. and F.Š.; funding acquisition, F.Š. and C.M. All authors have read and agreed to the published version of the manuscript.

**Funding:** This project was supported by the Specific Research Project of University of Hradec Kralove (No. 2102/2023) (P.B.) and by the Ministry of Education, Youth, and Sports of the Czech Republic (SVV 260 686/2023) (J.K.). The authors acknowledge the Viničná Microscopy Core Facility (VMCF of

the Faculty of Science, Charles University), an institution supported by the MEYS CR (LM2023050 Czech-BioImaging), for their support and assistance in this work. Fieldwork on the islands of Helgoland and Langeoog was funded by the German Red List Centre (D/396/67312935).

**Data Availability Statement:** The data presented in the study are available in the article and in public databases.

**Acknowledgments:** We are greatly indebted to Gerald Legg and Lutz Lange/collaborators, who kindly collected fresh material for DNA sequencing from England and Saxony-Anhalt, respectively. We acknowledge the uncomplicated loan of material from the following museum curators/staff: Jason Dunlop/Anja Friedrichs (ZMHB), Christoph Hörweg (NHMW), and Paolo Glerean (MFSN). Special thanks are due to Giulio Gardini (Genoa) for the loan of comprehensive material from his collection, the supply of literature, help with translation and various information and advice. Luis Guariento (Padova) kindly provided access to handwritten notes by Canestrini, and he delivered images of the type specimen of *Chelifer ninnii* Canestrini from the collection at Zoology Museum of Padova University. Jörg Spelda supplied literature that would otherwise have been difficult to access. C.M. thanks H.-P. Reike for the demonstration of the usefulness of flower pot coasters for the collection of pseudoscorpions. We thank four reviewers for the positive evaluation and their encouragement.

**Conflicts of Interest:** The authors declare no conflicts of interest.

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
