# Peer review of "And Yet They Differ: Reconsiderations of Diversity within Dactylochelifer latreillii (Arachnida: Pseudoscorpiones)"

_diversity, doi:10.3390/d16030137_

Round 1
Reviewer 1 Report
Comments and Suggestions for Authors
Well structured, results clearly presented, good discussion!
Only a few typos (see file attached), please use the official abbreviation for the Natural History Museum Vienna, which is NHMW (to be changed in TableS2 as well).

Author Response
Thank you very much for the positive and encouraging review report, and for spotting the typos. We changed NMW to NHMW and corrected the typos.
Reviewer 2 Report
Comments and Suggestions for Authors
please correct line 274 "Villaggio" instead of "Villagio", and so in Table S3 lines 191 to 229.
Author Response
Thank you very much for the positive and encouraging review report. We corrected Villagio del Pescatore to Villaggio del Pescatore in text and Table S3.
Reviewer 3 Report
Comments and Suggestions for Authors
It is an excellent article, I believe that by using different tools to differentiate the species we can see more clearly which characters help us distinguish the species. With what you found, is it possible to design a key to separate the Dactylochelifer species?
Author Response
Thank you very much for the positive and encouraging review report. Right now we don't think it makes much sense to prepare a determination key, as it would contain just the characters from the diagnoses. As we deal only with three species, information for determination can easily be taken from the diagnoses.
Reviewer 4 Report
Comments and Suggestions for Authors
Please provide the current distribution of Dactylochelifer latreillii s. l. (the range on the base of the published records) and (if possible) briefly discuss it.
Author Response
Thank you very much for the positive and encouraging review report. Based on your suggestion, we included a sentence on the distribution of Dactylochelifer latreillii s. l. at the end of the manuscript